# The Mediating Role of Work Satisfaction in the Relationship between Organizational Communication Satisfaction and Organizational Commitment of Healthcare Professionals: A Cross-Sectional Study

**DOI:** 10.3390/healthcare11060806

**Published:** 2023-03-09

**Authors:** Helmy Jameel Abu Dalal, Vimala Ramoo, Mei Chan Chong, Mahmoud Danaee, Yousef I. Aljeesh, Veshaaliini Uma Rajeswaran

**Affiliations:** 1Department of Nursing Science, Faculty of Medicine, Universiti Malaya, Kuala Lumpur 50603, Malaysia; 2Department of Social and Preventive Medicine, Faculty of Medicine, Universiti Malaya, Kuala Lumpur 60503, Malaysia; 3Scientific Research and Postgraduate, Islamic University of Gaza, Gaza City 00972, Palestine; 4Department of Medicine, Universiti Kebangsaan Malaysia Medical Centre, Kuala Lumpur 56000, Malaysia

**Keywords:** healthcare professionals, work satisfaction, organizational communication, organizational commitment

## Abstract

The factors that influence the organizational commitment of healthcare professionals, particularly organizational communication and work satisfaction, are essential for all healthcare organizations. This is particularly important for those who are under the pressure of high demand, economic constraints, and staff shortages. This study examined the relationship between organizational communication satisfaction and organizational commitment among healthcare professionals and the mediating role of work satisfaction in the relationship. A validated self-administered questionnaire and a universal sampling approach were used to conduct a cross-sectional survey of 235 healthcare professionals in the Gaza Strip, Palestine. The data were analyzed using SPSS version 25 and SmartPLS software to perform partial least squares structural equation modeling (PLS-SEM). A STROBE checklist was used to report the results. The results show a significant positive correlation between all measured variables. Work satisfaction partially mediates the relationship between organizational communication satisfaction and organizational commitment. The results of the PLS-SEM analyses suggest that communication satisfaction and work satisfaction account for 61% of the variation in organizational commitment. This study reveals that work satisfaction and communication satisfaction are imperative to building a sense of commitment in healthcare professionals. These results reinforce the existing evidence on the factors that influence the organizational commitment of healthcare professionals. Strategies to better shape internal communication practices and improve the work environment through regular feedback to healthcare professionals are essential to strengthening their organizational commitment.

## 1. Introduction

Healthcare professionals (HCPs) are the cornerstone in the provision of safe, efficient, and high-quality care; thus, organizations need to have a highly qualified and experienced workforce. In this regard, organizational commitment is critical to maintaining a high retention rate of HCPs [1]. Organizational commitment is defined as “a mindset or psychological state concerning the employee’s relationship with an organization” [1]. It characterizes the relationship between an organization and its members. Organizational commitment is demonstrated through members’ acceptance of the organization’s values (identification), willingness to exert effort on behalf of the organization (involvement), and decision to stay with or leave the organization (loyalty) [1,2]. An employee will be committed to their organization when their identity is linked to the organization and when the organization’s and the employee’s goals are aligned [3]. According to Meyer and Allen, commitment has three states: affective, normative, and continuance [1]. Affective commitment occurs when an employee “wants to stay” with the organization; normative commitment occurs when an employee feels they “ought to stay” with the organization; and continuance commitment occurs when an employee believes they “need to stay” with the organization [1]. There is considerable interest in organizational commitment because highly committed employees are theorized to exhibit more positive citizenship behavior and job performance and have a higher retention rate [3,4,5].

The literature has revealed that various factors influence employees’ organizational commitment, including work satisfaction [5,6,7], burnout [8], work relationships [9], and organizational communication [4]. Employees’ satisfaction with communication and work are among the top factors affecting organizational commitment [8]. For example, Bell and Sheridan’s cross-sectional study of 756 nurses in Ireland revealed a positive relationship between organizational commitment and work satisfaction [3]. Similarly, Geun and Park’s cross-sectional study of 239 nurses in three teaching hospitals in Korea demonstrated higher organizational commitment and productivity among highly satisfied nurses [4]. 

Organizational communication is an integral part of the work process within an organization. It refers to how information and ideas are exchanged between employees and management in an organization [10]. In addition, it is through communication that an employee learns their work process, what is expected of them, and how superiors judge their work [11,12]. Therefore, in a work environment, the quality of intra-organizational communication and employees’ perceived satisfaction with it play an important role in influencing work satisfaction at different levels and with different types of employees [7]. Organizational communication failure between HCPs and their managers leads to frustration, stress, and impaired relationships [13]. In addition, organizational communication satisfaction (OCS) is associated with greater work engagement [14], a crucial factor driving nurses’ retention [8]. Despite its importance, communication is frequently overlooked in the HCP context [15]. In particular, the impact of OCS on HCPs’ organizational commitment and work satisfaction has been underexplored globally. 

Work satisfaction is employees’ level of cognition and affection toward their work, which provides the foundation for their work attitudes [16]. Classically, work satisfaction is defined as “a pleasurable positive emotional state resulting from the appraisal of one’s job or job experiences” [17]. Most scholars agree that work satisfaction is an emotional response to different aspects of work [18]. The degree of employees’ work satisfaction or dissatisfaction influences their reaction to work and has a significant effect on their organizational commitment [3,19]; thus, a high level of work satisfaction is a critical condition for work or organizational commitment. Work satisfaction among HCPs is a crucial element in providing safe and effective health services [20] and can affect their satisfaction with the internal communication practices, which, in turn, could affect their commitment to work and the organization. It is implied here that improving communication in health organizations can enhance HCPs’ work satisfaction and decrease their turnover rate, which is associated with high costs. The 2021 NSI National Health Care Retention and RN Staffing Report highlighted the NSI Nursing Solution study that found that the average turnover cost of a bedside registered nurse in the United States ranges from $33,300 to $56,000, resulting in an average loss of $3.6 million to $6.1 million for the hospitals [21]. This could create an enormous strain on healthcare organizations on top of the existing strains related to the pressures of high demand, global economic constraints, and staffing shortages.

According to social exchange theory, various resources such as knowledge, information, feelings, attitude, mutual respect, and understanding can be exchanged during communication. Therefore, a positive and fair exchange of communication between HCPs and their managers can create a positive emotional response to various aspects of work and foster a sense of being challenged and a sense of enthusiasm, inspiration, and pride in one’s work, subsequently consolidating organizational commitment among HCPs through acceptance of the organization’s values and a willingness to exert effort on behalf of the organization [3,4,8]. Work satisfaction appears to be an essential resource for increasing work commitment; satisfied HCPs experience positive emotions such as happiness, joy, and enthusiasm [7,11]. Hence, OCS as the primary exchange and work satisfaction as a secondary exchange can lead to commitment as a positive outcome or obligations between HCPs and their organization.

The literature review noted that there is a paucity of empirical research on the relationship between organizational communication and organizational commitment in HCPs and that the impact of work satisfaction between the variables has not been widely considered in healthcare organizations. In addition, previous studies have only tested the direct effects of the variables, and most studies have been conducted in other parts of the world. Therefore, measuring work satisfaction and organizational commitment among Palestinian HCPs is crucial, as the HCPs in this country face many challenges, including restrictions, uncertainty, insecurity, overwork, stress, and a lack of specialists and medical resources, in addition to political and economic instabilities [22,23]. In such situations, communication processes within healthcare organizations are likely to be compromised, and work satisfaction and organizational commitment may be low. Therefore, it is important to assess the organizational and personal factors that can influence HCPs’ commitment to their organization in order to improve the quality of healthcare services in the Palestinian context. 

The primary aim of this study was to determine the relationship between OCS, organizational commitment, and work satisfaction and to examine the mediating effect of work satisfaction on the relationship between OCS and organizational commitment in PRCS HCPs. The following hypotheses are proposed and tested:
There are significant relationships between OCS, work satisfaction, and organizational commitment, and the sub-hypotheses (H1a–c) were:-There is a significant relationship between OCS and work satisfaction (H1a)-There is a significant relationship between OCS and organizational commitment (H1b)-There is a significant relationship between work satisfaction and organizational commitment (H1c)Work satisfaction mediates the relationship between OCS and organizational commitment (H2).

## 2. Methods

### 2.1. Research Design, Setting, and Sample 

A cross-sectional design is adopted using a self-administered survey questionnaire. The target respondents are HCPs (i.e., nurses, physicians, and paramedics) providing primary, secondary, and tertiary healthcare services under the Palestinian Red Crescent Society (PRCS). This approach was used in this study because the data collection included a large group of HCPs and only a single time point. Furthermore, a descriptive cross-sectional design is suitable for establishing statistically significant associations between the variables in this study [24].

The Palestinian RCS is a leading humanitarian organization affiliated with the International Red Cross and Red Crescent Movement and has been providing medical services to Palestinians for more than 50 years. Sample adequacy is determined using a 5% significance level, 80% statistical power, an *R^2^* value of 0.25, and the number of arrows pointing at a latent variable; 59 responses are considered adequate [24]. However, prior research suggests that a path analysis requires a sample size of 100 to 200 as a good starting point [25,26]. Therefore, considering the possible attrition rates, the total population of 300 HCPs from two tertiary hospitals, four primary healthcare centers, and five emergency healthcare centers who met the research criteria were invited to participate in the study using a total population sampling method known as universal sampling, a type of non-probability sampling technique that examines the entire population [24].

### 2.2. Study Constructs and Measures

A self-administrated questionnaire with four parts was used; the first part related to information on respondents’ sociodemographic details (i.e., gender, education level, job position, tenure of service, employment status, and income level), while the remaining three related to the variables measured in this study (i.e., OCS, organizational commitment, and work satisfaction). Prior studies suggest that the demographic variables influence the study variables [3,4,7]; therefore, they are included in the analysis.

#### 2.2.1. Organizational Communication Satisfaction (OCS)

The Communication Satisfaction Questionnaire (CSQ) proposed by Downs and Hazen is adopted to measure OCS [27]. The scale has 35 items classified into the following three dimensions: 15 items on interpersonal communication (IPC), 15 items on quality of organizational communication (QOC), and five items on coworker communication (COC). An example of an item representing the QOC subscale is “Information about organization policies and goals.” All items are scored on a seven-point Likert scale, ranging from 1 for “very dissatisfied” to 7 for “very satisfied.”

#### 2.2.2. Organizational Commitment

Organizational commitment is assessed using the Organizational Commitment Questionnaire (OCQ). The 15-item instrument measures commitment in two dimensions: value commitment (10 items) and commitment to stay (5 items). An example of an item representing the value commitment subscale is “I am willing to put in a great deal of effort beyond that normally expected in order to help this organization be successful.” The items are measured on a 7-point Likert scale, ranging from 1 (never) to 7 (always).

#### 2.2.3. Work Satisfaction

The Minnesota Satisfaction Questionnaire (MSQ) is used to measure work satisfaction. It has 20 items under two subdomains: intrinsic factors (IN) with 12 items and extrinsic factors (EX) with 8 items. An example of an item representing the EX subscale is “The feeling of accomplishment I get from the job.” Similar to the other variables, a 7-point Likert scale, ranging from 1 for “very dissatisfied” to 7 for “very satisfied,” is used to measure the items.

The original English versions of the scales were adopted after receiving approval from their authors. Six bilingual experts translated the items into Arabic following the recommendations of Tsang et al. for “forward-backward” translation [28]. Another six experts from different medical fields and with different levels of experience (e.g., medicine, nursing, public health, and health management) evaluated the accuracy, clarity, comprehensibility, and comprehensiveness of the items. Based on the experts’ feedback, the variables were found to have satisfactory item-content validity index scores with values ranging from 0.83 to 1.00 and reasonable Kappa coefficient values ranging from 0.82 to 1.00. Some items were slightly modified to improve their sentence structure based on the experts’ suggestions. A psychometric pretest of the Arabic instrument with 200 samples was carried out prior to the actual data collection, and the findings showed the internal consistency coefficients of the scales for OCS, organizational commitment, and work satisfaction are 0.95, 0.91, and 0.93, respectively. For the sub-dimensions of each scale, the coefficients range from 0.83 to 0.95, demonstrating superior internal consistency. These findings are consistent with previous internal consistency results of the English versions of the OCS [10], OCQ [29], and MSQ [30].

### 2.3. Data Collection and Ethical Considerations

Data were collected from January to April 2019 after securing ethical clearance from the university’s Institutional Review Board (MRECID No. 2018122-5979) and the local Health Research Council (PHRC/HR/350/18), in addition to written approval from the participating study institutions. The principal researcher collected data physically with the assistance of designated staff from each institution. The questionnaires were given to the participants in a sealed envelope after obtaining their written consent. The respondents had one week to complete the questionnaires, and mobile messages were sent as a reminder. This study, fully in line with the Declaration of Helsinki, assured participants of confidentiality and anonymity. The reporting of this study follows the guidelines for reporting observational studies (Appendix A) in the statement on Strengthening the Reporting of Observational Studies in Epidemiology (STROBE).

### 2.4. Data Analysis

The data were checked for completeness and distribution before inferential analysis using SPSS software (version 25). SmartPLS version 3.0 software is used to perform partial least squares structural equation modeling (PLS-SEM) [31,32]. This approach involves testing the research hypotheses in two stages. First, the measurement model is evaluated for convergence and discriminant validity, and then the convergent validity of the constructs is measured by testing the average variance extracted (AVE). Hair et al. suggested that standardized outer loadings should be equal to or greater than 0.70 and the recommended level of average variance extracted (AVE) be above 0.5; the lowest acceptable level of composite reliability (CR) is 0.7 [26,31]. Three methods were used to test discriminant validity: namely, the Hetrotrait-Monotrait ratio of correlations (HTMT criterion), the Fornell-Larcker criterion, and the cross-loading criterion. HTMT was applied in this study to assess discriminant validity across all model constructs. An HTMT value less than 1 indicates that the constructs in the model are far away from one another [31,32]. According to Fornell and Larcker (1981), adequately discriminant validity is reached when the square root of AVE is greater than the correlations between the latent variables [33]. Substantially, the cross-loading criterion was used to test whether any distinctive indicators or items that are hypothesized to measure a particular construct are closely related to another different construct in the model and do not measure other things. Item loadings in the main factor are higher than loadings in other factors, and a difference of 0.2 between loadings is accepted [26]. Then, the structural model was assessed to ascertain the path coefficients using bootstrapping. Finally, a mediation analysis was performed to test the indirect effect. The assessment of mediation includes testing for total effects as the sum of the direct and indirect effects using the symbols *c* = *c’ + ab* [26]. A significance level of 5% is set for all the tests.

## 3. Results

### 3.1. Respondents’ Sociodemographic Profile

A total of 235 respondents participated in this study with a 78.3% response rate. The details of the respondents’ background characteristics are described in Table 1.

### 3.2. Level of Organizational Commitment, OCS, and Work Satisfaction

Descriptive statistics were initially used to assess the level of OCS, work satisfaction, and organizational commitment by computing the mean (*M*) and standard deviation (*SD*) of each dimension and the overall construct. All variables were measured using a 7-point Likert scale; high scores expressed high levels of agreement among participants. OCS had an overall mean score of 4.96 (*SD* = 0.94), indicating moderate communication satisfaction among HCPs; the highest satisfaction level is found in communication among coworkers, and quality of communication is the least satisfying aspect. The overall mean score of work satisfaction is 5.11 (*SD* = 0.97), indicating participants’ moderate satisfaction with their work. Moreover, HCPs were more satisfied with the intrinsic factors (*M* = 5.42; *SD* = 0.94) surrounding their work than they were with the extrinsic factors (*M* = 4.80; *SD* = 1.19) surrounding it. The overall mean score of organizational commitment is 4.67 (*SD* = 0.98), showing that HCPs had moderate commitment to their organization. In addition, the mean value for the first sub-dimension, “value commitment” (*M* = 5.24; *SD* = 1.12), is higher than that of the second sub-dimension, “commitment to stay” (*M* = 4.09; *SD* = 1.29).

### 3.3. Relationship between Demographic Characteristics and the Variables

To determine the influence of the demographic characteristics on the variables (i.e., organizational commitment, OCS, and work satisfaction), univariate analyses are performed using an independent t-test and an ANOVA test. HCPs with 5 to 10 years of experience have a significantly higher OCS than HCPs with less than 5 years or more than 10 years of experience do (*p* = 0.010). Moreover, managers (*M* = 5.20; *SD* = 1.03) have a significantly higher (*p* < 0.05) level of organizational commitment than non-managers do (*M* = 4.60; *SD* = 0.95). There is no statistically significant variation in HCPs’ mean work satisfaction score across sociodemographic characteristics. The details of the results of the analyses of the relationship between the demographic characteristics and the variables are presented in Table 1.

### 3.4. Relationships among OCS, Work Satisfaction, and Organizational Commitment

The results of the Pearson’s bivariate correlation analysis reveal that OCS and work satisfaction are strongly and positively correlated [*r* (233) = 0.76; *p* < 0.001]. However, organizational commitment correlates more positively with work satisfaction [*r* (233) = 0.60; *p* < 0.001] than it does with OCS [*r* (233) = 0.48; *p* < 0.001] (Table 2).

### 3.5. Model Testing

#### 3.5.1. First-Order Model

The first stage of PLS-SEM analysis is the assessment of the measurement model. The findings demonstrate that the convergent validity of the variables is within acceptable ranges for the items’ external loads, composite reliability (CR), and extracted mean variance (AVE). Only one organizational obligation item (item CS2) had an outer loading of less than 0.5 and has therefore been deleted because it did not contribute to the obligation to stay dimension (Table 3).

The HTMT criterion recommended by Henseler et al. is used to assess discriminant validity [32]. The HTMT values of all variables are less than 0.90, ranging from 0.33 to 0.84. Thus, the variables have ample discriminant validity. The Fornell–Larcker criterion is used to determine discriminant validity by making a comparison between the square root of the AVE of each variable and its correlation with the other variables in the model. The highest value of the squared correlation between the sub-dimensions is 0.76. Following the principle of Fornell and Larcker, this finding implies that the measurement model has adequate discriminant validity [33]. In addition, the Arabic versions of OCSQ, MSQ, and OCQ have a high level of internal consistency, with a Cronbach’s α of 0.97, 0.94, and 0.90, respectively.

#### 3.5.2. Second-Order Model

The bootstrapping results for the second-order models are presented in Table 4. The findings show that the three sub-dimensions of OCS, IPC (*β* = 0.94; *p* < 0.001), Quality Communication (QC) (*β* = 0.94; *p* < 0.001), and COC (*β* = 0.75; *p* < 0.001), contribute significantly to OCS as a second-order latent variable. Likewise, the two dimensions of organizational commitment—commitment to stay (*β* = 0.53; *p* < 0.001) and value commitment (*β* = 0.99; *p* < 0.001)—contribute significantly to OCS as a second-order latent variable. Finally, the two sub-dimensions—extrinsic factors (*β* = 0.95; *p* < 0.001) and intrinsic factors (*β* = 0.86; *p* < 0.001)—significantly contribute to work satisfaction as a second-order latent variable. The standardized path coefficients (outer loadings) for all paths are above 0.7 and significant.

The relationships between the variables (path coefficients and significance) and the indicators of model adjustment, i.e., predictive relevance (*Q_2_*), Pearson’s coefficient of determination (*R_2_*), and effect size (*f_2_*), are measured in the structural model. The squared multiple correlation (*R_2_*) value of the model’s dependent variable, organizational commitment, is 0.61, revealing that 61% of the variance in organizational commitment is explained by OCS and work satisfaction (Figure 1). Moreover, the adjusted *R_2_* for work satisfaction is 0.60, implying that 60% of the variance in work satisfaction can be attributed to OCS. Based on Cohen’s effect size guideline [34], the effect of OCS on organizational commitment is small (*f_2_* = 0.04), while the effect of OCS on work satisfaction is large (*f_2_* = 1.54) [34]. The results also show that the *Q_2_* value of work satisfaction (*Q_2_* = 0.29) is greater than zero, suggesting that OCS affects work satisfaction. Likewise, the *Q_2_* value of organizational commitment is greater than zero (*Q_2_* = 0.30), which indicates that OCS, as an independent variable, and work satisfaction, as a mediator variable, have a significant effect on organizational commitment. As depicted in Figure 1 and Table 5, the standardized path coefficients (*β*), the significance of the paths (*p*-value), and the *R_2_* of each endogenous variable are tested. The hypothesis that *β* > 0 is accepted at the 5% significance level.

According to the results of the hypothesis testing, the effect of OCS on work satisfaction is positive and statistically significant (*β* = 0.78; *p* < 0.001), supporting H1a. The relationship between OCS and organizational commitment is also significant and positive (*β* = 0.23; *p* = 0.003), supporting H1b. Likewise, the impact of work satisfaction on organizational commitment is positive and statistically significant (*β* = 0.26; *p* = 0.032), supporting H1c. The total effect is *β* = 0.42 [direct effect (*β*c’ = 0.23) + mediation effect (*βab* = 0.20)]. The result shows that work satisfaction increases the effect of OCS on organizational commitment from *β* = 0.23 to *β* = 0.44, supporting H2 in that work satisfaction partially mediates the relationship between OCS and organizational commitment.

## 4. Discussion

Organizational commitment is among the most vital components for boosting employee morale and work productivity, which can improve an organization’s ability to achieve its goals. Therefore, creating an organizational culture that emphasizes commitment, especially in healthcare, is a major concern of human resource management worldwide. Factors such as work satisfaction and satisfaction with an organization’s communication practices are reported to affect employee commitment. Academic interest in all the variables measured in this study has steadily increased. Therefore, this study focuses on exploring whether the organizational commitment of HCPs is influenced by OCS and work satisfaction and examines the role of work satisfaction in the relationship between OCS and organizational commitment. A hierarchical multicomponent research model is proposed to test the research hypotheses. Overall, the findings of this study provide empirical support for the hypothesized relationships between these variables.

The first three hypotheses of this study postulated a positive relationship between the variables. The results of this study support previous studies and show a significant relationship between OCS and work satisfaction [7,30], OCS and organizational commitment [4,6], and work satisfaction and organizational commitment [3,18,35]. This means that satisfied HCPs are more committed to their organization. This result indicates that management should focus on improving internal communication to foster positive relationships and optimal commitment levels among HCPs. This study finds that 61% of the variance in organizational commitment is accounted for by OCS and work satisfaction, indicating that HCPs’ satisfaction with their organization’s communication and work is essential for their organizational commitment.

The fourth hypothesis suggests that work satisfaction mediates the relationship between OCS and organizational commitment. This prediction is also correct, and the results reveal that work satisfaction partially mediates the relationship between the two variables. Work satisfaction has a significant impact on the relationship between OCS and organizational commitment. This finding elucidates that high levels of work satisfaction positively reinforce and solidify the relationship between OCS and organizational commitment in HCPs [30,36]. Therefore, creating a supportive work culture and improving work conditions, such as benefits, level of autonomy and freedom at work, perceived professional development, and relationships with co-workers and superiors, can increase work commitment and thus provide a retention strategy for HCPs. A study reported that dissatisfied nurses were 2.5 times more likely to leave the profession [37]. 

Since this study finds that HCPs are more satisfied with intrinsic factors than they are with extrinsic factors, attention should be paid to work-related aspects, such as autonomy, opportunities, training, support, and guidance, to meet HCPs’ needs. Palestinian health organizations face dynamic changes, such as the increasing disease burden, financial and resource constraints, and political instability; these challenges have led to job insecurity among HCPs. Therefore, health care managers must constantly evaluate their HCPs’ work satisfaction levels and the factors that affect their level of satisfaction. In addition to intrinsic needs, the quality of physical structures (e.g., patient areas, safety, and workspaces) and the work environment (supervisor support and openness to communication) are among the key factors that managers should consider in order to increase HCP commitment [16].

The finding that HCPs’ demographic characteristics influence OCS and organizational commitment is consistent with most studies. Vermeir et al. (2018) found a statistically significant difference in OCS based on years of experience [36]. Furthermore, Ahmad and Oranye (2010) found organizational commitment at the managerial level to differ significantly between nurses [35]. However, in this study, HCPs’ work satisfaction is not influenced by their demographic characteristics, suggesting that HCPs have similar needs and expectations from their work factors regardless of differences in their personal and professional characteristics.

## 5. Study Limitations

Although this study is the first to draw significant conclusions about organizational communication between Palestinian HCPs, it has a few limitations. First and most importantly, the cross-sectional design of this study limits the causality of the findings. The direction of the relationships between the variables remains open, although a significant assumption is made about the directions. Therefore, a longitudinal or experimental study is needed to address this issue. Second, this study was conducted in a non-governmental organization; it is recommended that future studies include diverse governmental and non-governmental healthcare organizations for comparison and better generalization of the findings to all HCPs. Third, the study deals exclusively with the Palestinian context, which is faced with political unrest and economic restrictions, and is therefore limited in its generalizations to other contexts. It is recommended that future studies survey other countries to understand the impact of differences in socioeconomic status, culture, and healthcare systems on the variables.

## 6. Conclusions

The findings of this study contribute to the literature on the antecedents of organizational commitment by confirming the positive relationship between OCS and work satisfaction with organizational commitment, although the causal direction is not examined. This study suggests that communication and work satisfaction have the most substantial effect on organizational commitment. A reasonable level of work satisfaction is important for HCPs to translate their OCS into organizational commitment. The findings show that as HCPs’ work satisfaction increases, their satisfaction with their organizational communication also increases. Thus, they are likely to exhibit greater organizational commitment and be less willing to leave the organization. Therefore, faced with the difficulties in recruiting newly graduated HCPs and the negative impact of losing highly skilled HCPs, Palestinian health policymakers should develop strategies to retain existing HCPs by improving their work environment and communication processes.

Because of work satisfaction’s strong contribution to greater organizational commitment, managers should consider strategies for enhancing work satisfaction by investing substantial resources. In addition, the regular identification of the needs of HCPs and the measures required to meet these needs should be a priority. As corporate communications cement the organization’s commitment and consensus, action is needed in order to emphasize effective communication strategies, including trusting and supportive communications. In particular, clear and open communication regarding an organization’s policies, goals, achievements, failures, and financial condition should be provided on a consistent and regular basis to all levels of HCPs.

## Figures and Tables

**Figure 1 healthcare-11-00806-f001:**
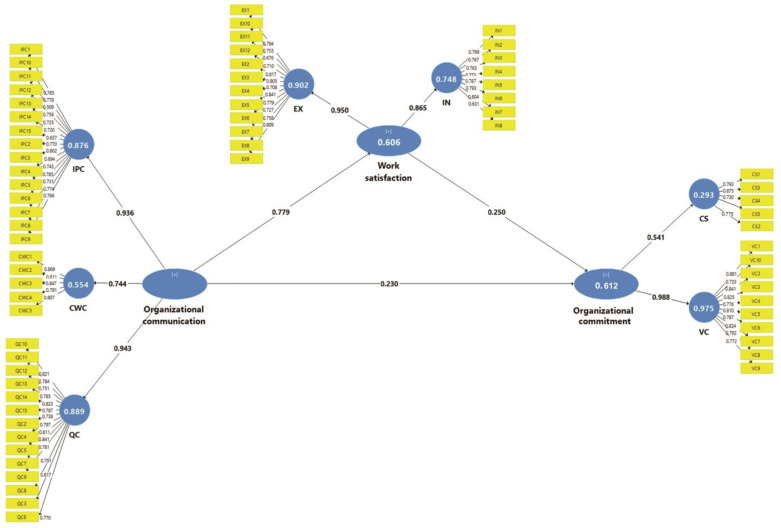
PLS algorithm on mediating the effect of work satisfaction on the relationship between organizational communication satisfaction and organizational commitment. Note: IPC: interpersonal communication; QC: quality communication; COC: coworker communication; CS: commitment to stay; VC: value commitment; EX: extrinsic factor; IN: intrinsic factor.

**Table 1 healthcare-11-00806-t001:** HCPs’ organizational communication satisfaction and organizational commitment according to demographic characteristics (N = 235).

Demographic	*n*	OCS (*M ± SD*)	Statistical Value	*p*	OC (*M± SD*)	Statistical Value	*p*	WS (*M ± SD*)	Statistical Value	*p*
Gender			0.30 ^t^	0.764		0.79 ^t^	0.430		0.06 ^t^	0.949
Men	146	4.97 ± 0.95			4.71 ± 0.98			5.11 (1.00)		
Women	89	4.93 ± 0.93			4.73 ± 0.98			5.12 (0.94)		
Years of experience			7.55 ^a^	0.001 *		1.26 ^a^	0.285		1.81 ^a^	0.166
Less than 5 years	68	5.08 ± 0.76			4.54 ± 1.06			5.00 (099)		
5–10 years	77	5.21 ± 0.80 ^d^			4.65 ± 0.86			5.29 (0.91)		
More than 10 years	90	4.65 ± 1.08 ^c^			4.79 ± 1.01			5.05 (1.02)		
Salary			2.35 ^a^	0.076		0.81 ^a^	0.488			
Less than RM 2000	36	4.99 ± 0.80			4.56 ± 0.73			4.91 (1.03)	2.35 ^a^	0.076
RM 2000–2500	53	5.16 ± 0.67			4.65 ± 0.91			5.18 (0.76)		
RM 2501–3000	52	5.03 ± 1.02			4.85 ± 1.00			5.39 (0.92)		
More than RM 3000	94	4.78 ± 1.05			4.63 ± 1.09			5.01 (1.07)		
Job position			0.14 ^t^	0.886		3.12 ^t^	0.002 *		0.48 ^t^	0.634
Non-manager	205	4.96 ± 0.93			4.60 ± 0.95			5.10 (0.96)		
Manager	30	4.93 ± 1.03			5.20 ± 1.03			5.19 (1.09)		
Job category			0.26 ^a^	0.768		0.42 ^a^	0.655		0.73 ^a^	0.481
Physician	76	4.99 ± 1.00			4.61 ± 1.07			5.09 ± 1.08		
Nurse	115	5.00 ± 1.00			4.67 ± 0.83			5.07 ± 0.93		
Paramedic	44	5.12 ± 1.05			4.78 ± 1.17			5.28 ± 0.86		
Education			0.05 ^a^	0.953		0.74 ^a^	0.529		0.70 ^a^	0.550
Diploma	20	4.93 ± 1.14			4.73 ± 0.82			5.15 ± 0.91		
Bachelor’s	176	4.97 ± 0.94			5.53 ± 1.00			5.12 ± 0.98		
PhD & Master’s	39	4.93 ± 0.89			5.78 ± 0.88			5.20 ± 1.4		

Note: OCS: organizational communication satisfaction; OC: organizational commitment; WS: work satisfaction. * Significant, *p* < 0.05; *M =* mean; *SD* = standard deviation. ^t^ Analyzed based on an independent *t*-test; ^a^ Analyzed based on a one-way ANOVA; ^d^ Compared 5–10 vs. more than 10; ^c^ Compared more than 10 vs. less than 5 years.

**Table 2 healthcare-11-00806-t002:** Correlation between OCS, work satisfaction, and organizational commitment.

	*r* (*p*)		
Variables	1	2	3
1. OCS	1		
2. WS	0.76 (<0.001)	1	
3. OC	0.48 (<0.001)	0.60 (<0.001)	1

Note: OCS: organizational communication satisfaction; WS: work satisfaction; OC: organizational commitment.

**Table 3 healthcare-11-00806-t003:** Descriptive statistics on study variables.

Variables	Mean	*SD*	No. of Items	Factor Loading Range	Cronbach’s Alpha	AVE	CR
Interpersonal communication	5.01	1.00	14	0.625–0.807	0.940	0.562	0.947
Quality communication (QC)	4.72	1.12	15	0.742–0.844	0.960	0.643	0.964
Co-worker communication (COC)	5.13	0.98	5	0.755–0.869	0.883	0.676	0.912
Value commitment (VC)	5.24	1.12	10	0.733–0.881	0.939	0.648	0.948
Commitment to stay (CS)	4.09	1.29	4	0.746–0.857	0.726	0.535	0.815
Extrinsic satisfaction (EX)	5.11	0.97	12	0.676–0.841	0.936	0.587	0.944
Intrinsic satisfaction (IN)	5.42	0.94	8	0.631–0.804	0.896	0.587	0.917

Note: CR: composite reliability; AVE: average variance extracted; *SD*: standard deviation.

**Table 4 healthcare-11-00806-t004:** Test of second-order models using bootstrapping.

Variables	Outer Loading	*SE*	*T* Value	*p*-Value
Organizational communication → COC	0.75	0.03	22.60	<0.001
Organizational communication → IPC	0.94	0.01	69.13	<0.001
Organizational communication → QC	0.94	0.01	132.72	<0.001
Organizational commitment → CS	0.53	0.07	7.76	<0.001
Organizational commitment → VC	0.99	0.01	312.87	<0.001
Work satisfaction → EX	0.95	0.01	138.79	<0.001
Work satisfaction → IN	0.86	0.02	37.20	<0.001

Note: IPC: interpersonal communication; QC: quality communication; COC: coworker communication; CS: commitment to stay; VC: value commitment; EX: extrinsic factor; IN: intrinsic factor; *p*: statistical significance; *SE*: standard error; *β*: standardized coefficient beta.

**Table 5 healthcare-11-00806-t005:** Summary of path coefficients and hypotheses testing (mediating results).

Hypothesis	Path	*Β*	*SE*	*T* Value	*p*-Value
Direct effect	OCS → Organizational commitment	0.23	0.07	3.01	0.003 *
Indirect effect	OCS → WS→ OC	0.20	0.09	2.15	0.031
Total effect	OCS → Organizational commitment	0.42	0.03	27.78	<0.001 **

Note: * Significant, *p* < 0.05; ** Significant, *p* < 0.001; IPC: interpersonal communication;QC: quality communication; COC: coworker communication; *p*: statistical significance;SE: standard error; *β*: standardized coefficient beta.

## Data Availability

Data available on request from the authors.

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
