# Peer review of "The Mediating Role of Work Satisfaction in the Relationship between Organizational Communication Satisfaction and Organizational Commitment of Healthcare Professionals: A Cross-Sectional Study"

_healthcare, 2023, doi:10.3390/healthcare11060806_

Round 1
Reviewer 1 Report
Overall, I really enjoyed the research and work done on this paper as the information is well organised and presented in a coherent way.
Nevertheless, I detected a small flaw in terms of methodology. Firstly, on page 3 they talk about objectives and then they present hypotheses. The authors should define the research objectives and properly enunciate all the hypotheses created for the research. This is because regarding the hypotheses, on page 3, the authors present two hypotheses, but in the discussion of the results they talk about four hypotheses.
In fact, although the final results are clear, this methodological flaw concerning the hypotheses should be corrected so that the scientificity of the article is not called into question.
Reviewer 2 Report
Abstract
- “The factors influencing the organizational commitment of healthcare professionals (…) are (…)”.
- When the authors address the objectives of their study, they should use the past tense: “This study examined (…)”.
- “The data was analyzed (…)”.
- This study followed a cross-sectional design. As such, the authors should not use expressions that point to the existence of a cause-effect relationship (e.g., impact).
- I advise the authors to reread this section to improve the language presented.
Introduction
- The authors should avoid presenting statistical results in the Introduction (e.g., r=0.59; p<.001). It is enough to give an overview of the results of the cited studies.
- Throughout this section, the authors present the selected constructs for their research. Also, they cite some studies that have explored the association between the different variables. However, they do not explain how and why those constructs are related, i.e., what are the underlying mechanisms that explain the relationship in question.
- Also, what does it mean the “XXX” in line 101 of the Introduction?
- At the end of this section, the authors should present a more descriptive explanation of the importance and pertinence of their study and how it can contribute to theory and practice.
Objectives
- What is the meaning of “XXX” in line 112?
- The authors present the main aim of their study; this is correct, but what about the specific objectives?
- Also, the authors should have presented the research hypotheses more descriptively.
Methods
- The authors should consult, e.g., the American Psychological Association (APA) manual, to understand that different verb tenses should be used in different parts of the manuscript.
- The authors should present references to support the arguments about the required sample size. In the literature, there is no agreement regarding this aspect. The definition of sample sizes typically follows a rule of thumb. Thus, the authors should cite sources supporting the arguments regarding the sample size.
- The information regarding the reliability of the instruments should be placed in the text, near the presentation of the different measures.
- Also, the authors should have presented an example item for each measure used.
- Data analysis section should be more descriptive, e.g., in the Results section, the authors have used the Average Variance Extracted (AVE) and Composite Reliability (CR) coefficients and the Fornell and Larcker criterion, but they did not mention them in this section. For example, what are these coefficients' purpose, and what are their cut-off values?
Results
- Statistical symbols must be in italics (e.g., SD in line 209).
Discussion
- The authors mentioned that their hypotheses were confirmed. However, they did not explain what this confirmation means.
- Also, they should develop more on their results' theoretical and practical implications.
Conclusion
- Nothing to add.
Author Response
The reply to Reviewer 2 is as attached.

Author Response
The reply to the review is in the attached file.
